# Exploring the Mechanistic Interplay between Gut Microbiota and Precocious Puberty: A Narrative Review

**DOI:** 10.3390/microorganisms12020323

**Published:** 2024-02-04

**Authors:** Min Yue, Lei Zhang

**Affiliations:** 1Microbiome-X, National Institute of Health Data Science of China & Institute for Medical Dataology, Cheeloo College of Medicine, Shandong University, Jinan 250012, China; 2Department of Biostatistics, School of Public Health, Cheeloo College of Medicine, Shandong University, Jinan 250012, China; 3State Key Laboratory of Microbial Technology, Shandong University, Qingdao 266237, China

**Keywords:** precocious puberty, gut microbiota, endocrine disruption, metabolism and obesity, children’s health, microbiota–gut–brain axis, probiotics

## Abstract

The gut microbiota has been implicated in the context of sexual maturation during puberty, with discernible differences in its composition before and after this critical developmental stage. Notably, there has been a global rise in the prevalence of precocious puberty in recent years, particularly among girls, where approximately 90% of central precocious puberty cases lack a clearly identifiable cause. While a link between precocious puberty and the gut microbiota has been observed, the precise causality and underlying mechanisms remain elusive. This narrative review aims to systematically elucidate the potential mechanisms that underlie the intricate relationship between the gut microbiota and precocious puberty. Potential avenues of exploration include investigating the impact of the gut microbiota on endocrine function, particularly in the regulation of hormones, such as gonadotropin-releasing hormone (GnRH), luteinizing hormone (LH), and follicle-stimulating hormone (FSH). Additionally, this review will delve into the intricate interplay between the gut microbiome, metabolism, and obesity, considering the known association between obesity and precocious puberty. This review will also explore how the microbiome’s involvement in nutrient metabolism could impact precocious puberty. Finally, attention is given to the microbiota’s ability to produce neurotransmitters and neuroactive compounds, potentially influencing the central nervous system components involved in regulating puberty. By exploring these mechanisms, this narrative review seeks to identify unexplored targets and emerging directions in understanding the role of the gut microbiome in relation to precocious puberty. The ultimate goal is to provide valuable insights for the development of non-invasive diagnostic methods and innovative therapeutic strategies for precocious puberty in the future, such as specific probiotic therapy.

## 1. Introduction

It is well known that the human gut microbiota undergoes a transitional phase around weaning and subsequently stabilizes [1], evolving towards an adult-like composition within 2–3 years after birth [2,3,4]. However, differences in microbial community structure and diversity persist between children and adults [5,6]. Meanwhile, the gut microbiota (GM) begins to exhibit gender-specific differences during adolescence, suggesting a link between GM and sexual maturation [3,5]. Interestingly, disruptions in GM accompany abnormal sexual maturation, with the highest incidence observed in precocious puberty [7].

Precocious puberty refers to the onset of secondary sexual characteristics in girls before the age of 8 and in boys before the age of 9 [8]. Central precocious puberty (CPP) is the most common type of precocious puberty, characterized by the premature activation of the hypothalamic–pituitary–gonadal axis (HPGA), 15–20-times more prevalent in girls than boys [9]. This condition not only affects the adult height of patients but also increases the risk of various diseases in childhood and adulthood [10]. Currently, despite the identification of some causes of CPP, up to 90% of girls with CPP have no clear explanation for the premature activation of their HPGA [9,11,12]. Additionally, the CPP diagnosis and treatment methods still result in the poor acceptance of children, often accompanied by adverse reactions and other problems. Therefore, there is an urgent need to further explore the etiology of precocious puberty to provide clues for the development of non-invasive diagnostic and therapeutic methods.

Recent research has indicated associations between precocious puberty and the gut microbiota and its metabolites [7,13,14,15,16,17]. However, the specific mechanisms remain unclear. The purpose of this narrative review is to summarize the potential mechanisms of interaction between the gut microbiota and precocious puberty, including metabolic pathways, hormonal regulation, nutritional status, and the gut–brain axis, to provide insights into the discovery of new targets for precocious puberty and the development of microbiota-based interventions (probiotics, prebiotics, and fecal microbiota transplantation) for its management.

## 2. Methods

A comprehensive literature assessment was conducted through a systematic search of the PubMed database, spanning from its inception to December 2023. The review encompassed original articles, meta-analyses, reviews, and animal studies, focusing on the intricate interplay between puberty and gut microbiota. The search employed specific terms, including “gut microbiota”, “polycystic ovary syndrome”, “puberty”, “precocious puberty”, “central precocious puberty”, “peripheral precocious puberty”, “hormones”, “probiotics”, “hypothalamic-pituitary-gonadal axis”, “metabolites”, “fecal microbiota transplantation”, and “obesity”. The inclusion of articles in this review was meticulously determined by the authors based on their relevance to constructing a cohesive narrative review.

## 3. Gut Microbiota

### 3.1. Developmental Trajectory of Gut Microbiota

The gut microbiota, as a complex microbial ecosystem within the human body [18], undergoes evolutionary changes at different stages of human life, potentially impacting health [19].

Early life is crucial for the establishment of the gut microbiota [20,21,22]. During this period, the gut microbiota evolves from a relatively simple structure into a more complex and stable community [23,24,25]. Factors, such as mode of delivery [23,26,27,28,29,30,31], breastfeeding [28,31,32], early antibiotic exposure [3,24,28,31,33,34], and host-related factors [34,35,36], play decisive roles in the early development of the gut microbiota.

The gut microbiota in childhood and adolescence begins to exhibit a certain degree of stability and complexity [1,4,37]. However, compared to adulthood, the gut microbiota in children may be more susceptible to environmental influences [38] and has not yet reached the complexity of the adult microbiota [19,39]. Importantly, gender differences in the gut microbiome emerge during adolescence and persist into adulthood [40]. A cross-sectional study revealed that the gut microbiota’s diversity remains stable in pre-adolescent subjects after the age of 5, with no apparent gender differences. However, upon entering adolescence, significant differences in the gut microbiota between adolescent males and females become evident, suggesting a close association between adolescent sexual maturation and changes in the gut microbiota [41]. This study unveiled gender-specific differences in the gut microbiota during different adolescent stages, with gender-dependent gut microbiota diversity linked to sex hormones [41]. Nonetheless, dynamic changes in the gut microbiota before and after adolescence are still lacking in research [41]. Furthermore, the specific mechanisms of interaction between hormonal changes during adolescence and the gut microbiota require further investigation.

In adulthood, the gut microbiota typically reaches a relatively stable state but remains influenced by dietary habits, lifestyle, medication use (especially antibiotics), and health status [42]. With advancing age, the gut microbiota in elderly individuals may undergo changes due to alterations in dietary habits, immune decline, chronic diseases, and medication use [18,43].

### 3.2. Functions of Gut Microbiota

The gut microbiota exerts profound effects on the health and disease status of the host, encompassing functions related to nutrient metabolism, host immune regulation, and its potential impact on the central nervous system (Figure 1) [44,45].

The gut microbiota plays a crucial role in nutrient metabolism [46]. Gut bacteria are involved in the breakdown of dietary fibers [47] and undigested proteins within the host [45], generating beneficial metabolites, such as short-chain fatty acids (SCFAs) [48] (e.g., butyrate and propionate), neuroactive compounds (e.g., nitric oxide), polyamines, and aromatic compounds. SCFAs, in particular, not only provide energy for intestinal epithelial cells [49] but also modulate host energy balance [50,51], regulate inflammatory immune responses [51,52], and enhance intestinal barrier function [48,53]. Furthermore, the breakdown metabolism also produces γ-aminobutyric acid (GABA), norepinephrine, dopamine, histamine, and serotonin [54], which have an impact on regulating the gut–brain axis or maintaining host nitrogen balance [45].

The gut microbiota can also communicate with the central nervous system via the gut–brain axis and participate in the regulation of various central nervous system-related disorders [55]. The gut and brain form the gut–brain axis through bidirectional neural, endocrine, and immune communication, where changes in one organ can affect the other [56,57,58]. The gut microbiota composition and its secreted metabolic products and signaling molecules can influence both the enteric nervous system and the central nervous system, thereby affecting host emotions and behavior [59]. The symbiotic gut microbiota is closely associated with various central nervous system disorders and psychiatric illnesses [60], such as Parkinson’s disease [61,62,63], Alzheimer’s disease [64,65,66,67], schizophrenia [68,69], multiple sclerosis [70], depression [71,72,73,74], autism [75,76], and anxiety disorders [55,77,78]. This discovery provides a new perspective for the treatment of central nervous system disorders and psychiatric illnesses.

## 4. Precocious Puberty

### 4.1. Definition

Precocious puberty is typically defined as the onset of signs of puberty in girls before the age of 8 and in boys before the age of 9 [8]. Precocious puberty can be categorized into two main types based on its underlying mechanisms: CPP and peripheral precocious puberty. CPP results from the premature activation of the HPGA, which is the most common type. Peripheral precocious puberty, on the other hand, is caused by abnormal hormone secretion without involving the premature activation of the HPGA [79].

### 4.2. Epidemiology of Precocious Puberty

Precocious puberty demonstrates a global trend of prevalence [80], but the incidence rates vary across different countries and regions. An observational study in the United States showed that by the age of 7, 10% of white girls and 23% of black girls had entered puberty [81]. In Europe, approximately 5% of girls start developing breast tissue before the age of 8 [82]. A study in Denmark reported an incidence rate of 0.2% for girls and less than 0.05% for boys [83]. These differences may be related to factors, such as race, lifestyle, and environmental factors. Additionally, precocious puberty is more common in girls than in boys, with up to 90% of female patients having idiopathic precocious puberty, meaning the cause is unknown [9,11,12,84,85]. Precocious puberty in boys is less common and often associated with organic lesions such as hypothalamic tumors [85,86].

### 4.3. Etiology and Risks of Precocious Puberty

Although some causes of precocious puberty have been identified, including congenital and acquired central nervous system damage, genetic mutations, obesity [87,88,89], exposure to endocrine-disrupting chemicals, and early exposure to sex hormones, many of the mechanisms behind idiopathic precocious puberty remain unclear [9]. Meanwhile, precocious puberty has significant impacts on the physical and psychological health of sufferers. It not only affects their adult height [90] and psychological well-being [91] but also increases the risk of various diseases during adulthood, including mental disorders, hypertension, type II diabetes [92], ischemic heart disease [93], stroke, estrogen-dependent cancers [94], cardiovascular diseases, and more [9]. Therefore, a further exploration of the mechanisms underlying precocious puberty in girls is urgently needed to provide a theoretical basis for the development of prevention, diagnosis, and treatment strategies for precocious puberty.

### 4.4. Current Diagnosis and Treatment of Precocious Puberty

The key to diagnosing precocious puberty in children lies in distinguishing between CPP and peripheral precocious puberty, with the gonadotropin-releasing hormone (GnRH) stimulation test being the gold standard for this differentiation [95,96]. This test aims to determine if the pituitary gland can be stimulated by GnRH and release gonadotropins, such as follicle-stimulating hormone (FSH) and luteinizing hormone (LH), to stimulate the gonads (ovaries or testes) to produce sex hormones (estrogen or testosterone) [95,96]. In this test, a GnRH analogue is injected into the child, and the peak levels of LH and the LH/FSH ratio in the child’s blood within 120 min after the injection are observed. If the LH peak is >5 IU/L [95,97] and the LH/FSH ratio is >0.6 [98], a diagnosis of CPP is made. However, this test can be painful and distressing for the child, leading to poor acceptance by both the child and their family. Therefore, there is an urgent need to develop new non-invasive diagnostic strategies.

Currently, treatment options for children with precocious puberty also face challenges related to poor acceptance by the children and the occurrence of various adverse reactions. For peripheral precocious puberty or slowly progressing precocious puberty in children, treatment often involves dietary and lifestyle adjustments. On the other hand, for rapidly progressing CPP, treatment often requires the administration of GnRH agonists [9]. The principle behind this treatment is that GnRH analogues bind to the GnRH receptors on the anterior pituitary gonadotroph cells, preventing them from binding to GnRH secreted by the hypothalamus, thus suppressing the pituitary–gonadal axis. This leads to a reduction in LH and FSH release, feedback inhibition of sex hormone secretion from the ovaries, a decrease in testosterone levels, and delayed skeletal maturation [99,100]. However, this treatment approach also presents several challenges, including the pain and poor compliance associated with repeated injections, high medical costs, potential adverse reactions (such as allergies and vaginal bleeding), and local complications like aseptic abscesses that can lead to treatment inefficacy [101,102]. Therefore, optimizing the current treatment strategies for precocious puberty is urgently needed to improve children’s acceptance and minimize adverse reactions. Future research should focus on developing safer, more effective, and easily accepted treatment methods, including the development of new drugs and personalized treatment plans. Additionally, providing psychological support and education for patients and their families is crucial to enhance the overall effectiveness of precocious puberty treatment and improve the quality of life for sufferers.

### 4.5. Dysbiosis of Gut Microbiota in Precocious Puberty

In recent years, increasing evidence has suggested that dysbiosis of the gut microbiota may play a significant role in the pathogenesis of precocious puberty in children. An observational study in China included 27 girls with CPP, 24 overweight girls, and 22 healthy controls, revealing differences in gut microbiota among the three groups [7]. Among them, girls with CPP showed increased alpha diversity in their gut microbiota, significant increases in bacteria, such as *Alistipes*, *Klebsiella*, and *Sutterella*, and enhanced inter-bacterial correlations [7]. Another cross-sectional study demonstrated significant differences in gut microbiota and metabolites between children with CPP and healthy controls, with the Streptococcus genus potentially serving as a candidate molecular marker for CPP treatment [103]. Similarly, in an observational study, fecal samples from 25 girls with idiopathic CPP and 23 healthy girls were subjected to 16S rDNA sequencing to compare microbial compositions between the groups. The study found a significant enrichment in various bacterial species in girls with CPP, including *Ruminococcus gnavus*, *Ruminococcus callidus*, *Ruminococcus bromii*, *Roseburia inulinivorans*, *Coprococcus eutactus*, *Clostridium letum*, and *Clostridium lacatifermentans* [17]. These studies collectively demonstrate significant alterations in the structure and composition of gut microbiota in girls with precocious puberty, indicating an association between gut microbiota and precocious puberty. However, most studies are based on 16S rDNA or 16S rRNA sequencing techniques, which may not be sufficient to detect small microbiota alterations compared to metagenomics sequencing [104]. Furthermore, the causal relationship and underlying mechanisms between gut microbiota and precocious puberty require further exploration. A deeper investigation into the role and mechanisms of gut microbiota in the development of precocious puberty could provide new insights into the pathogenesis of precocious puberty and the development of novel non-invasive diagnostic and therapeutic strategies.

## 5. Interaction and Potential Mechanisms between Gut Microbiota and Precocious Puberty

Figure 2 depicts the factors involved in the interactions between gut microbiota and precocious puberty. These factors are discussed in greater detail in the following paragraphs.

### 5.1. Metabolic Pathways

The gut microbiota is considered a metabolic “organ” that not only aids in extracting nutrients and energy from ingested food but also produces a myriad of metabolites that regulate the host’s metabolism through their homologous receptors [105]. The interaction between the gut microbiota, its metabolites, and host metabolic pathways plays a critical role in host health and homeostasis. Simultaneously, several metabolic pathways within the bodies of children with precocious puberty undergo changes, including alterations in lipid metabolism, bile acid metabolism, amino acid metabolic pathways, and neurotransmitter metabolic pathways.

#### 5.1.1. Neurotransmitter Metabolic Pathways

In a study by Li et al., a comparison of gut microbiota between 27 girls with CPP and 22 healthy controls revealed the enrichment of *Alistipes*, *Klebsiella*, and *Sutterella* in CPP patients’ gut microbiota [7]. These bacteria, often abundant in patients with neurological disorders [106], may trigger the early onset of puberty by secreting metabolites related to neurotransmission, such as serotonin and dopamine, which activate the HPGA [7]. This suggests that specific gut microbiota may influence precocious puberty by secreting neurotransmitter-like metabolites, acting on the HPGA. However, further validation and exploration of potential mechanisms are needed.

#### 5.1.2. Amino Acid Metabolic Pathways

Moreover, several observational studies have shown associations between nitric oxide synthesis and the progression of precocious puberty [103]. Functional predictions of the gut microbiota in girls with CPP show increased nitric oxide synthesis and positive correlation with FSH and insulin in children with CPP [7]. However, whether nitric oxide exacerbates precocious puberty and its underlying mechanisms requires further investigation.

#### 5.1.3. Lipid Metabolism

Currently, the most studied metabolites regarding their impact on precocious puberty are SCFAs, which are derived from the gut microbiota. Several observational studies have pointed to the synthesis and metabolism of SCFAs as enriched metabolic pathways in the gut microbiota of CPP, peripheral precocious puberty PPP, and healthy children [17]. These studies demonstrate significant differences in SCFA-related metabolic products between children with precocious puberty and healthy children. However, the specific mechanisms and causative relationships remain unclear. Nevertheless, an animal study conducted by Wang et al. provided some preliminary answers to this question. Wang et al. [15] investigated the effects of gut microbiota and its derived SCFAs on the HPGA in obese-induced precocious puberty rats by adding acetate, propionate, butyrate, or their mixture to a high-fat diet. The study found that obese-induced precocious puberty rats experienced an earlier first estrous cycle, increased expression of *Kiss1*, *GPR54*, and *GnRH* in the hypothalamus, and early gonadal maturation. Additionally, the gut microbiota of precocious puberty rats exhibited dysbiosis, and the production of SCFAs was reduced. Adding acetate, propionate, butyrate, or their mixture to the high-fat diet significantly reversed precocious puberty in rats, reduced hypothalamic GnRH release, and delayed the development of the gonadal axis through the *Kiss1-GPR54-*PKC-ERK1/2 pathway. This study provides the first causal-level evidence that gut microbiota-derived SCFAs can reverse the process of sexual maturation and partially elucidates the underlying mechanisms. This finding offers important clues for the non-invasive treatment of precocious puberty, demonstrating that gut microbiota-derived SCFAs are a promising therapeutic approach for preventing obesity-induced precocious puberty.

#### 5.1.4. Bile Acid Metabolism

Bile acids are another essential metabolic product of the gut microbiota, originating from endogenous molecules synthesized from cholesterol in the liver and further metabolized by the gut microbiota [107]. Existing research has confirmed that glycodeoxycholic acid induces the secretion of interleukin-22 (IL-22) by group 3 innate lymphoid cells in the intestine, and IL-22 subsequently improves polycystic ovary syndrome (PCOS) [108]. Some studies suggest that CPP and PCOS share a common pathogenic basis: dysfunction of the HPGA [109]. However, research on the relationship between bile acids and precocious puberty is still lacking. Therefore, investigating the influence of bile acids on precocious puberty and the associated mechanisms is a promising direction, providing a theoretical basis for developing new targets for the non-invasive treatment of precocious puberty.

### 5.2. Hormonal Regulation

Precocious puberty is a condition closely associated with sex hormones. Premature exposure to high levels of gonadotropins, including LH and FSH, triggers the secretion of sex hormones from the gonads, advancing sexual maturity and leading to a series of diseases related to prolonged exposure to high levels of sex hormones. *Gemmiger* and LH enriched in the gut microbiota of girls with precocious puberty show a positive correlation, as well as *Fusobacterium* and FSH [17]. In a high-fat-diet-induced precocious puberty mouse model, GnRH was positively correlated with *Desulfovibrio*, *Lachnoclostridium*, *gCA-900066575*, *Streptococcus*, *anaerobic bacteria*, and *Bifidobacterium* [14]. This suggests that gut microbiota abundance is related to hormone levels, implying that the gut microbiota may influence precocious puberty by regulating hormone levels. Conversely, hormone levels may also affect the abundance of certain gut microbiota. Koren et al. observed significant changes in the gut microbiota composition during pregnancy in 91 women, irrespective of their health status, particularly during the three months of pregnancy when estrogen levels peak [110,111]. The gut microbiota and its metabolites are associated with host insulin sensitivity [17]. Insulin can induce the adrenal secretion of androgens and regulate LH pulse secretion [112,113], indirectly indicating an association between the gut microbiota and sex steroid hormones.

Microbial-secreted beta-glucuronidase can metabolize estrogen from its conjugated form to its deconjugated form. Dysbiosis and reduced diversity in the gut microbiota decrease beta-glucuronidase activity, leading to decreased estrogen deconjugation, reduced circulating estrogen, and resulting in highly pathogenic conditions, such as obesity, metabolic syndrome, cardiovascular diseases, and cognitive decline [114,115]. An increase in the number of bacteria producing beta-glucuronidase can elevate circulating estrogen levels and lead to diseases such as endometriosis and cancer [116]. These studies collectively demonstrate that the gut microbiota is closely related to various sex hormone-related diseases, including precocious puberty, complications of pregnancy, adverse pregnancy outcomes, PCOS [117], endometriosis, and cancer. However, mechanistic research in this area is limited, and more effort should be directed towards exploring the potential pathogenic mechanisms mediated by the microbiota–hormone axis, offering novel therapeutic and preventive strategies.

### 5.3. Nutritional Status

A healthy nutritional status is crucial for the growth and puberty development of children and adolescents [118,119,120], and it is estimated to account for up to 25% of the variation in pubertal timing [118]. An increasing body of research indicates that the timing of puberty onset is associated with overnutrition and an energy imbalance. An animal experiment by Wang et al. [16] found that offspring mice exposed to high-fat diets during maternal lactation exhibited signs of adolescent obesity, early puberty, irregular estrous cycles, and glucose metabolism disorders. Co-housing these offspring mice with those from mothers on a normal diet reversed changes in gut microbiota composition, early puberty, and insulin insensitivity induced by maternal high-fat diets. This suggests that early-life high-fat diets may lead to precocious puberty and that the colonization of a healthy gut microbiota can reverse this phenomenon, further indicating the involvement of obesity-related gut microbiota in early puberty [16]. In another animal experiment, post-weaning high-fat diets were found to increase serum estradiol, leptin, deoxycholic acid (dCA), and GnRH in mice, leading to precocious puberty [14]. Population studies have also shown the phenomenon of gut microbiota involvement in precocious puberty in connection with obesity, with gut microbiota enriched in girls with idiopathic CPP showing a correlation with obesity [17].

Furthermore, early nutrition is a key factor in puberty development. Breast milk appears to play a critical role in puberty development, as it is a unique source of beneficial bacteria and compounds that shape the infant’s microbiota and influence the development of various physiological functions, such as the gastrointestinal, immune, and nervous systems [121]. Breastfeeding directly exposes infants to maternal gut microbiota through the gut–breast axis and indirectly shapes the infant’s gut microbiota during early life [121], further affecting the process of puberty development. Hvidt et al., based on a cohort study, found that a lack of breastfeeding was associated with accelerated puberty development in boys, while there was no significant correlation with girls’ puberty development [122]. Al-Sahab et al. observed a negative correlation between exclusive breastfeeding and age at menarche in a population-based cohort study [123]. Similarly, Ong et al. demonstrated in the Avon Longitudinal Study of Parents and Children (ALSPAC) cohort that breastfeeding might have a protective effect on early menarche in girls [124]. These studies emphasize the complex and sometimes inconsistent role of breastfeeding in influencing the timing of puberty, but the specific mechanisms remain unclear.

Therefore, nutritional status plays an important role in the occurrence and development of precocious puberty, not only by affecting body weight and hormone levels but also through indirect pathways, such as influencing the gut microbiota. In-depth research into the role of nutritional intake in precocious puberty will help better understand the mechanisms underlying precocious puberty and may provide valuable information for the development of prevention and intervention strategies.

### 5.4. The Potential Role of the Gut Microbiota–Brain Axis in Precocious Puberty

Scientists proposed that the gut microbiota can influence brain function and behavior through various mechanisms, including the microbiota’s immune response, metabolism, neurotransmitters, and the gut–brain neural pathway [125]. The concept of the microbiota–gut–brain axis has provided a new perspective for studying the pathogenesis of brain diseases, identifying new intervention targets, and holds promise for the concept of “treating brain diseases through the gut.” In the pathogenesis of CPP, the premature activation of the HPGA is a critical component. Therefore, it is essential to investigate the overall effect of the gut microbiota on the HPGA.

Existing research has found that antibiotic-induced alterations in the colonic microbiota of piglets regulate the expression of aromatic amino acids and neurotransmitters in the hypothalamus [126]. The gut microbiota regulates hypothalamic inflammation and leptin sensitivity in mice fed a Western diet through a GLP-1R-dependent mechanism [127]. These studies suggest that the gut microbiota can modulate the expression of metabolites and neurotransmitters in the hypothalamus, as well as gene expression levels.

The gut microbiota can also impact the gonadal organs within the HPGA. The gut microbiota and its metabolic products play a regulatory role in ovarian dysfunction and insulin resistance associated with PCOS. PCOS patients have significantly increased levels of common Bacteroides in their gut microbiota, while levels of glycodeoxycholic acid and tauroursodeoxycholic acid are reduced. Mechanistically, glycodeoxycholic acid induces the secretion of IL-22 by intestinal group 3 innate lymphoid cells through GATA binding protein 3, thereby improving the PCOS phenotype [108]. Interventional studies have shown that improving the content of beneficial gut bacteria can promote spermatogenesis. Gut microbiota transplantation can increase sperm concentration by 2–3-times and sperm vitality by approximately 10-times. At the cellular level, there is a gradual increase in the number of cells in the seminiferous tubules, from spermatogonia to spermatocytes, round sperms, and spermatozoa. Metabolomic and testicular metabolic profiling have shown that fecal microbiota transplantation can improve blood metabolic products, regulate the testicular microenvironment through the bloodstream, and promote spermatogenesis, improving sperm count and vitality. Gut microbiota transplantation is highly specific and does not affect all diseases in the body but specifically regulates spermatogenesis [128].

In summary, the regulatory role of the gut microbiota on both the hypothalamus and gonads suggests that the gut microbiota may influence precocious puberty by acting on the HPGA rather than individual organs. However, there is currently no systematic study elucidating the impact of the gut microbiota on the HPGA, and the specific mechanisms and potential therapeutic applications require further investigation. Understanding the interaction between the gut microbiota and the HPGA in precocious puberty is not only crucial for revealing its pathological processes but may also provide a biological basis for the development of new treatment strategies.

### 5.5. The Potential Confounding Factors That Could Influence the Observed Associations between Gut Microbiota and Precocious Puberty

The interplay between gut microbiota and puberty may vary significantly based on genetic, environmental, and lifestyle factors. The genetic background accounts for approximately 50–80% of the variability in pubertal onset and progression [129]. Certain ethnic groups, notably African American and Hispanic populations, exhibit an earlier onset of puberty attributed to genetic and nutritional factors [130]. Concurrently, a substantial body of research indicates that genetic factors can also influence the human gut microbiome [131,132,133]. Host genetic polymorphisms and/or mutations progressively alter the gut microbiota, intersecting with other environmental influences, leading to changes in the composition and function of the gut microbial community [134]. Environmental factors, including substances with the potential to interfere with the endocrine system (such as phthalates, dioxins, polybrominated biphenyls, and polychlorinated biphenyls), appear to play a role in influencing the timing of puberty [129,135,136]. In future research investigating the link between gut microbiota and precocious puberty, as well as its underlying mechanisms, it is crucial to meticulously address and control for the impact of these confounding variables.

## 6. The Prospects of Microbiota-Associated Therapies

### 6.1. Probiotics

Probiotics have been widely used in recent years for the prevention and adjunctive therapy of various diseases due to their non-invasive nature [137,138,139,140]. Regarding probiotic-related research in precocious puberty, there are currently only two animal experimental studies that have been reported. One study reported that probiotic treatment could reverse soy isoflavone (SI)-induced precocious puberty in female mice, possibly due to an increased production of SCFAs brought about by changes in the gut microbiota of the recipient mice [141]. The probiotics used in this study were commercial, consisting of live *Bifidobacterium longum*, *Lactobacillus bulgaricus*, and *Streptococcus thermophilus* [141]. Another animal experiment demonstrated that probiotic intake could reverse early maternal separation stress-induced precocious puberty in female rats, indicating that probiotic therapy restored the normal onset of puberty in rodents [142]. The probiotics used in this study were also commercial, with ingredients including 95% *Lactobacillus rhamnosus R0011* and 5% *Lactobacillus helveticus R0052* [142]. These studies all suggest that probiotic intake has a stabilizing effect on the timing of puberty onset in rodents, although the specific mechanisms need further exploration. Furthermore, the translation of the aforementioned animal study results to the human population requires additional clinical trial evidence. Future research efforts should prioritize more population-based intervention studies. Simultaneously, current research on probiotics for precocious puberty lacks specificity and future research should further investigate probiotics specifically related to precocious puberty for more targeted and effective treatment.

However, concerning probiotic therapy, the alterations in the gut microbiota it induces may be transient [143]. Several probiotic treatment trials have failed to observe significant changes in the gut microbiota, suggesting that individuals might require an extended treatment duration to achieve therapeutic effects. Future clinical trial research should place greater emphasis on post-intervention follow-ups to assess the optimal intervention duration for probiotic therapy.

### 6.2. Fecal Microbiota Transplantation (FMT)

FMT refers to the transfer of fecal material containing the gut microbiota from a healthy donor to a recipient with dysbiosis, using methods, such as colonic infusion, nasogastric, nasoenteric, or endoscopic approaches. The objective is to restore the normal diversity and functionality of the gut microbiota [144,145]. Currently recognized as an effective treatment for recurrent *Clostridium difficile* infection [146], FMT is also considered a potential therapy for certain extraintestinal diseases, including neurodegenerative disorders, owing to the bidirectional communication of the gut–brain axis [147,148]. Additionally, recent animal experimental evidence suggests FMT as a potential intervention for anti-aging and altering the life course stages [149]. This raises the question of whether FMT could also influence the timing of puberty, a crucial life stage, providing a novel direction for the treatment of precocious puberty. However, no studies have provided clues to date, and further research is warranted to explore the impact of FMT on the onset of puberty in humans. Table 1 summarizes data from studies on the interaction between gut microbiota and precocious puberty mentioned in the text. Table 2 summarizes the population studies mentioned in the text regarding the interaction between the gut microbiota and precocious puberty, along with an evaluation of the methodological limitations of the reviewed studies.

## 7. Discussion

The exact mechanisms underlying precocious puberty remain incompletely understood. Identified causes include congenital and acquired central nervous system damage, genetic changes, environmental endocrine disruptors, and premature exposure to sex hormones. However, the etiology of CPP remains unclear in 74–90% of affected girls, making it the most common form of precocious puberty [9,11,12]. The significant changes in the gut microbiota before and after puberty suggest a relationship with sexual maturation. Moreover, increasing observational studies indicate disturbances in the gut microbiota of children with precocious puberty. Non-invasive treatments, such as probiotics and fecal microbiota transplantation, are increasingly being used in clinical practice to adjunctively treat gastrointestinal diseases [150] (such as ulcerative colitis, Crohn’s disease, irritable bowel syndrome, etc.), brain–gut axis-related neurological diseases [151,152,153] (such as Parkinson’s disease, Alzheimer’s disease, anxiety, depression, autism, etc.), and endocrine system and immune system diseases [154,155] (such as diabetes, obesity, lupus, rheumatoid arthritis, allergies, etc.). This suggests that focusing on the mechanisms by which precocious puberty interacts with the gut microbiota is a promising research area, providing a theoretical basis for the development of specific probiotic strains or fecal microbiota transplantation treatments for precocious puberty.

However, it is currently unknown whether the gut microbiota can affect precocious puberty. Present studies predominantly adopt a cross-sectional approach, yielding correlations with limited persuasiveness. The necessity lies in conducting longitudinal cohort studies, offering more robust evidence with a causal direction. These studies are crucial to comprehensively investigate the dynamics of the gut microbiota and its potential involvement in the development of precocious puberty in humans. Additionally, animal experiments can validate whether the gut microbiota can regulate precocious puberty and elucidate the molecular mechanisms involved. Nevertheless, evidence from animal studies may not precisely replicate human physiology and pathology. Therefore, clinical trials are imperative to elucidate the causal association between the gut microbiota and precocious puberty in humans.

Building upon prior research, this paper suggests several novel avenues for investigating the interaction between gut microbiota and precocious puberty. First, the interaction between the gut microbiota and precocious puberty mainly occurs through four pathways: microbiota metabolism, microbiota–hormone interactions, microbiota nutritional status, and microbiota–HPGA interactions. These pathways are closely related, and future research should combine nutritional status, microbiota metabolism products, hormone regulation, and HPGA interactions to systematically elucidate the potential homeostatic system between the gut microbiota and precocious puberty. This will further enhance our understanding of the pathological mechanisms of precocious puberty. Secondly, probiotics and fecal microbiota transplantation (FMT) emerge as potential non-invasive treatments for precocious puberty. While the application of probiotics in treating precocious puberty is currently limited to animal experiments, we posit that future investigations should prioritize additional clinical experiments to further explore the therapeutic potential of probiotics for precocious puberty.

However, there are also limitations to this study. This narrative review, lacking specific criteria for evaluating the quality of population studies, has subjective inclusion and exclusion criteria for the literature. Therefore, in the future, there should be an effort to conduct systematic reviews in this field to the greatest extent possible.

## 8. Conclusions

In conclusion, the current research preliminarily confirms a correlation between precocious puberty in humans and the gut microbiota. Animal studies have shown that specific gut microbiota and their metabolites can reverse precocious puberty in rodent models. However, the causal effects and underlying interaction mechanisms between human precocious puberty and gut microbiota remain to be elucidated. This narrative review summarizes the potential molecular mechanisms mentioned in existing studies and proposes potential microbiome-related therapeutic approaches for precocious puberty. Future population studies are needed to clarify the causal relationship between the gut microbiota and human precocious puberty, as well as to elucidate their potential interaction mechanisms. Concurrently, clinical trials exploring specific probiotics and their metabolites for children with precocious puberty could provide new insights for non-invasive treatment options.

## Figures and Tables

**Figure 1 microorganisms-12-00323-f001:**
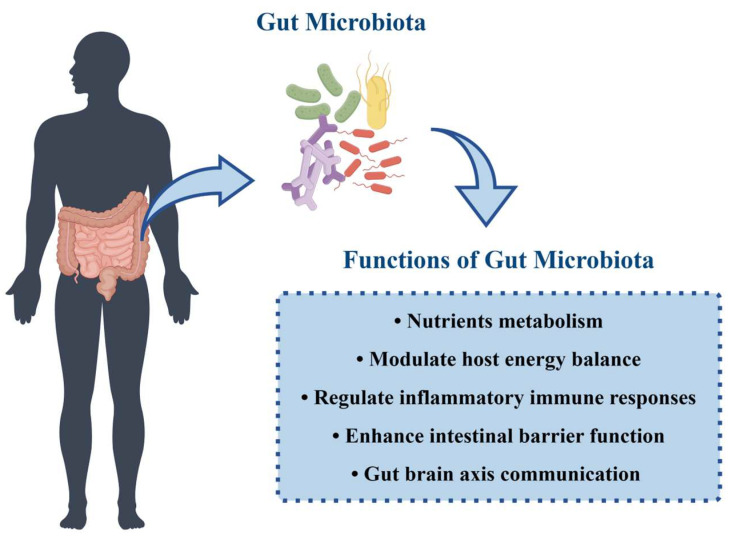
Functions of the gut microbiota. Figure was created by Figdraw (www.figdraw.com).

**Figure 2 microorganisms-12-00323-f002:**
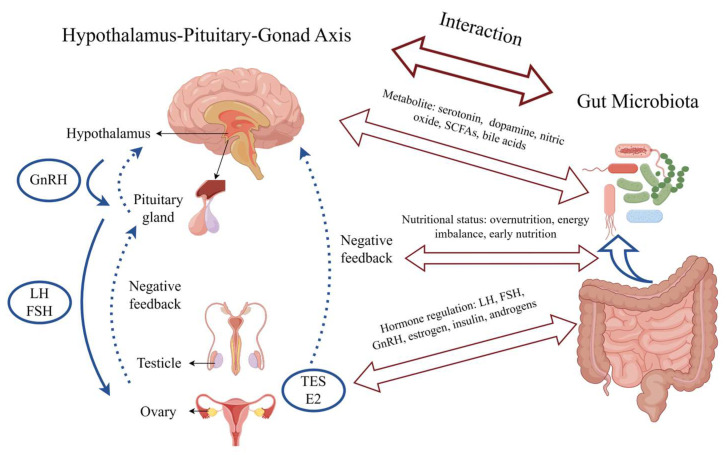
Interactions between gut microbiota and precocious puberty. Solid blue arrows represent positive feedback and dashed blue arrows represent negative feedback. Abbreviations: GnRH, gonadotropin-releasing hormone; LH, luteinizing hormone; FSH, follicle-stimulating hormone; TES, testosterone; E2, estradiol; SCFAs, short-chain fatty acids. Figure was created by Figdraw (www.figdraw.com).

**Table 1 microorganisms-12-00323-t001:** Current research on the interaction between gut microbiota and precocious puberty.

Author, Year	Location	Study Type	Sample Details	Methods	Key Findings
Li et al., 2021 [7]	China	Observational study	*n* = 73, 27 CPP girls, 24 over-weighted girls, 22 healthy controls	16S rRNA sequencing	1. The CPP group exhibited increased α-diversity in GM, significant elevations in *Alistipes*, *Klebsiella*, and *Sutterella* bacteria, enhanced inter-bacterial correlations 2. In the CPP group, increased nitric oxide synthesis positively correlated with FSH and insulin.
Dong et al., 2020 [17]	China	Observational study	*n* = 58, 25 ICPP girls, 23 healthy controls	16S rDNA sequencing	1. The ICPP group had higher GM diversity. 2. 36 candidate GM biomarkers for patients with ICPP screening were identified. 3. The GM of the ICPP group was enriched for the microbial functions of cell motility, signal transduction, and environmental adaptation. 4. Positive correlations were also detected between *Fusobacterium* and FSH, and *Gemmiger* and LH.
Wang et al., 2020 [16]	The United States	Animal experiment	C57BL/6 mice	16S rRNA sequencing	1. Co-housing reversed early puberty induced by MHFD during lactation through the fecal–oral route via increasing GM richness. 2. Co-housing reversed insulin insensitivity in offspring induced by MHFD during lactation.
Wang et al., 2022 [15]	China	Animal experiment	female Sprague Dawley rats	16S rDNA sequencing	SCFAs can act on *Kiss1* neurons and their receptor GPR54 and then reduce the release of hypothalamic GnRH and pituitary LH and FSH through the PKC-ERK1/2 pathway and delay the development of the ovary and uterus.
Bo et al., 2022 [14]	China	Animal experiment	C57 mice	16S rDNA sequencing, untargeted metabolomics sequencing	1. HFD after weaning caused precocious puberty in mice. 2 “HFD-microbiota” transplantation promoted the precocious puberty of mice. 3. Estrogen changes the composition and proportion of gut microbiota and promotes precocious puberty in mice.
Cowan et al., 2018 [142]	Australia	Animal experiment	Sprague Dawley-derived rats	Probiotic treatment	1. Stressed females exhibited earlier pubertal onset compared to standard-reared females, whereas stressed males matured later than their standard-reared counterparts. 2. A probiotic treatment restores the normative timing of puberty onset in rodents of both sexes.
Yuan et al., 2022 [141]	China	Animal experiment	female c57/bl mice	16S rRNA sequencing, Probiotic treatment	95% daidzein has the potential to advance the timing of puberty onset in female mice, and gut microbiome can be a therapeutic target to regulate the timing of puberty onset.
Huang et al., 2023 [103]	China	Observational study	*n* = 150, 91 CPP patients, 59 healthy controls	16S rRNA sequencing, untargeted metabolomics sequencing	1. Identified the altered microorganisms and metabolites during the development of CPP and constructed a classifier for distinguishing CPP. 2. Revealed the nitric oxide synthesis was closely associated with the progression of CPP and the genus *Streptococcus* could be a candidate molecular marker for CPP treatment.

Abbreviations: CPP, central precocious puberty; GM, gut microbiota; ICPP, idiopathic central precocious puberty; FSH, follicle-stimulating hormone; LH, luteinizing hormone; GnRH, gonadotropin-releasing hormone; MHFD, maternal high-fat diet; SCFAs, short-chain fatty acids; HFD, high-fat diet.

**Table 2 microorganisms-12-00323-t002:** Assessment of the quality of population studies on the interaction between the gut microbiota and precocious puberty.

Author, Year	Study Type	Sample Details	Methods	Key Findings	Methodological Limitations
Li et al., 2021 [7]	Observational study	*n* = 73, 27 CPP girls, 24 over-weighted girls, 22 healthy controls	16S rRNA sequencing	1. The CPP group exhibited increased α-diversity in GM, significant elevations in *Alistipes*, *Klebsiella*, and *Sutterella* bacteria, enhanced inter-bacterial correlations 2. In the CPP group, increased nitric oxide synthesis positively correlated with FSH and insulin.	16S rRNA sequencing provides taxa resolution up to the genus level and is unable to yield information on the functional characteristics compared to newer techniques such as shotgun-metagenome sequencing.
Dong et al., 2020 [17]	Observational study	*n* = 58, 25 ICPP girls, 23 healthy controls	16S rDNA sequencing	1. The ICPP group had higher GM diversity. 2. 36 candidate GM biomarkers for patients with ICPP screening were identified. 3. The GM of the ICPP group was enriched for the microbial functions of cell motility, signal transduction, and environmental adaptation. 4. Positive correlations were also detected between *Fusobacterium* and FSH, and *Gemmiger* and LH.	1. 16S rDNA cannot provide taxa information on the functional characteristics. 2. Fecal metabolomics were not investigated. 3. Researchers did not administer and analyze dietary questionnaires.
Huang et al., 2023 [103]	Observational study	*n* = 150, 91 CPP patients, 59 healthy controls	16S rRNA sequencing, untargeted metabolomics sequencing	1. Identified the altered microorganisms and metabolites during the development of CPP and constructed a classifier for distinguishing CPP. 2. Revealed the nitric oxide synthesis was closely associated with the progression of CPP and the genus *Streptococcus* could be a candidate molecular marker for CPP treatment.	1. Although 16 s rRNA sequencing was widely used to characterize microbial communities, it existed limitations in explaining complete genetic information compared to metagenomic sequencing. 2. Candidate microorganisms need to be further cultured to judge the origin of metabolites more accurately.

CPP, central precocious puberty; GM, gut microbiota; ICPP, idiopathic central precocious puberty; FSH, follicle-stimulating hormone; LH, luteinizing hormone.

## Data Availability

Not applicable.

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
