# Peer review of "Exploring the Mechanistic Interplay between Gut Microbiota and Precocious Puberty: A Narrative Review"

_microorganisms, 2024, doi:10.3390/microorganisms12020323_

Round 1
Reviewer 1 Report
Comments and Suggestions for Authors
Although the topic is interesting, the study results are admittedly difficult to interpret and potentially biased due to imbalanced weights of research findings and the authors' presuppositions.
Specific comments:
1. In the study title, it should be specified that this is a narrative review.
2. Figure 1 seems unnecessary especially since it is unclear what is the evidence base supporting the assertions.
3. The 16S rRNA sequencing method, which is used by most gut microbiota studies, may not be sensitive enough to detect small microbiota alterations (citation: pubmed.ncbi.nlm.nih.gov/37110143). This should be mentioned.
4. The paper does not specify the databases used, search terms, time frame of the literature search, or the inclusion and exclusion criteria for selecting studies. This makes it difficult for readers and other researchers reading the paper to assess the comprehensiveness and bias of the literature review. While it is acknowledged that there are no specific reporting guidelines for narrative reviews, it is expected that authors adhere to a systematic approach as far as possible.
5. Without assessing the quality of the individual studies referenced, the review risks basing its conclusions on potentially flawed or biased research, compromising the validity of its findings. Suggest appraising the human studies at least, making comments on the methodological limitations of the studies reviewed.
6. The authors have a tendency to generalize findings from animal studies to human populations. The review includes a significant number of animal studies without adequately addressing the limitations of extrapolating these findings to humans. Animal models, while informative, may not accurately replicate human physiology and pathology.
7. "In 2011, scientists proposed that the gut microbiota can influence brain function and behavior through various mechanisms, including the microbiota's immune response, metabolism, neurotransmitters, and the gut-brain neural pathway [124]" - I believe this was proposed way before 2011. The specific mention of the year 2011 is unnecessary.
8. The interplay between gut microbiota and puberty may vary significantly based on genetic, environmental, and lifestyle factors. The review does not sufficiently address potential confounding factors that could influence the observed associations between gut microbiota and precocious puberty.
9. The review includes a variety of study types (observational studies, animal experiments) which inherently have different levels of evidence. The authors could provide a more detailed discussion of how this heterogeneity affects the overall conclusions.
10. Regarding probiotics, the shift in the gut microbiota may be transient and temporary (citation: pubmed.ncbi.nlm.nih.gov/36986088) as several treatment trials for probiotics have failed to find significant alterations in gut microbiome; individuals may require longer duration of treatment to have therapeutic effects.
Comments on the Quality of English LanguageModerate edits required.
Author Response
Response to Reviewer 1
The authors would like to thank the reviewer for your helpful comments. We feel these comments have strengthened the manuscript considerably. The changes/revisions we made in the revised manuscript are highlighted in yellow color.
Reviewer #1:
Although the topic is interesting, the study results are admittedly difficult to interpret and potentially biased due to imbalanced weights of research findings and the authors' presuppositions.
Specific comments:
Comment 1: In the study title, it should be specified that this is a narrative review.
Response: Thank you for your detailed review. We have revised the title reads: “Exploring the Mechanistic Interplay Between Gut Microbiota and Precocious Puberty: A Narrative Review.” And we also added the word “narrative” on page 1 lines 17, page 1 line 28, and page 2 line59.
Comment 2: Figure 1 seems unnecessary especially since it is unclear what is the evidence base supporting the assertions.
Response: Thank you for your suggestion. Figure 1 summarizes the functions mentioned in the text of the gut microbiota so we remained it. In order to summarize the functions of the gut microbiota mentioned in the manuscript more accurately, we revised the label in the box describing gut microbiota functions reads: “Nutrients metabolism; Gut brain axis communication” (page 3).
Comment 3: The 16S rRNA sequencing method, which is used by most gut microbiota studies, may not be sensitive enough to detect small microbiota alterations (citation: pubmed.ncbi.nlm.nih.gov/37110143). This should be mentioned.
Response: Thank you for your detailed review. We added: “However, most studies are based on 16S rDNA or 16S rRNA sequencing techniques, which may not be sufficient to detect small microbiota alterations compared to meta-genomics sequencing.” and cited the study you mentioned (citation: pubmed.ncbi.nlm.nih.gov/37110143) on page 6, lines 235-238. In addition, we added Table 2 to evaluate the methodological limitations of the population studies reviewed. In the Table 2, we added: “16S rRNA sequencing provides taxa resolution up to the genus level and is unable to yield information on the functional characteristics compared to newer techniques such as shotgun-metagenome sequencing. 16S rDNA cannot provide taxa information on the functional characteristics. Although 16 s rRNA sequencing was widely used to characterize microbial communities, it existed limitations in explaining complete genetic information compared to meta-genomic sequencing.” (pages 13-14, Table 2).
Comment 4: The paper does not specify the databases used, search terms, time frame of the literature search, or the inclusion and exclusion criteria for selecting studies. This makes it difficult for readers and other researchers reading the paper to assess the comprehensiveness and bias of the literature review. While it is acknowledged that there are no specific reporting guidelines for narrative reviews, it is expected that authors adhere to a systematic approach as far as possible.
Response: Thank you for your review. We added the databases used, search terms, and the inclusion and exclusion criteria for selecting studies reads: “A comprehensive literature assessment was conducted through a systematic search of the PubMed database, spanning from its inception to December 2023. The review encompassed original articles, meta-analyses, reviews, and animal studies focusing on the intricate interplay between puberty and gut microbiota. The search employed specific terms, including "gut microbiota", "polycystic ovary syndrome", "puberty", "precocious puberty", "central precocious puberty", "peripheral precocious puberty", "hormones", "probiotics", "hypothalamic-pituitary-gonadal axis", "metabolites", “fecal microbiota transplantation”, and "obesity." The inclusion of articles in this review was meticulously determined by the authors based on their relevance to constructing a cohesive narrative review.” (page 2, lines 66-76). What’s more, we added the limitations in the conclusion reads: “However, we also realized the limitation of this study. This narrative review, lacking specific criteria for evaluating the quality of population studies, has subjective inclusion and exclusion criteria for literature. Therefore, in the future, there should be an effort to conduct systematic reviews in this field to the extent possible.” (page 15, lines 547-550).
Comment 5: Without assessing the quality of the individual studies referenced, the review risks basing its conclusions on potentially flawed or biased research, compromising the validity of its findings. Suggest appraising the human studies at least, making comments on the methodological limitations of the studies reviewed.
Response: Thank you for your suggestion. In accordance with the evaluation methods for assessing the quality of descriptive review studies, we added the Table 2 to assess the human studies associated with the interplay between the gut microbiota and precocious puberty and made comments on the methodological limitations of the studies reviewed (pages 13-14). Additionally, we added the limitation of the narrative review reads: “However, we also realized the limitation of this study. This narrative review, lacking specific criteria for evaluating the quality of population studies, has subjective inclusion and exclusion criteria for literature. Therefore, in the future, there should be an effort to conduct systematic reviews in this field to the extent possible.” for readers and other researchers reading the paper to assess the comprehensiveness and bias of the literature review (page 15, lines 547-550).
Comment 6: The authors have a tendency to generalize findings from animal studies to human populations. The review includes a significant number of animal studies without adequately addressing the limitations of extrapolating these findings to humans. Animal models, while informative, may not accurately replicate human physiology and pathology.
Response: Thank you for your detailed review. We added reads: “Furthermore, the translation of the aforementioned animal study results to the human population requires additional clinical trial evidence. Future research efforts should prioritize more population-based intervention studies.” (page 11, lines 463-465). Additionally, we revised in the section of conclusion reads: “Nevertheless, evidence from animal studies may not precisely replicate human physiology and pathology. Therefore, clinical trials are imperative to elucidate the causal association between the gut microbiota and precocious puberty in humans.” (page 15, lines 529-532).
Comment 7: "In 2011, scientists proposed that the gut microbiota can influence brain function and behavior through various mechanisms, including the microbiota's immune response, metabolism, neurotransmitters, and the gut-brain neural pathway [124]" - I believe this was proposed way before 2011. The specific mention of the year 2011 is unnecessary.
Response: Thank you for your suggestion. We revised the sentence as: “Scientists proposed that the gut microbiota can influence brain function and behavior through various mechanisms, including the microbiota's immune response, metabolism, neurotransmitters, and the gut-brain neural pathway.” (page 9, lines 393-395).
Comment 8: The interplay between gut microbiota and puberty may vary significantly based on genetic, environmental, and lifestyle factors. The review does not sufficiently address potential confounding factors that could influence the observed associations between gut microbiota and precocious puberty.
Response: Thank you for your review. We added the section “5.5. The Potential Confounding Factors that could influence the observed associations between gut microbiota and precocious puberty” to address the limitation. In this part, we summarized the confounding factors reported in previous study as: “The interplay between gut microbiota and puberty may vary significantly based on genetic, environmental, and lifestyle factors. The genetic background accounts for approximately 50-80% of the variability in pubertal onset and progression [129]. Certain ethnic groups, notably African American and Hispanic populations, exhibit an earlier onset of puberty attributed to genetic and nutritional factors [130]. Environmental factors, including substances with the potential to interfere with the endocrine system (such as phthalates, dioxins, polybrominated biphenyls, and polychlorinated biphenyls), appear to play a role in influencing the timing of puberty [129,131,132]. The interplay between gut microbiota and puberty may vary significantly based on genetic, environmental, and lifestyle factors. The genetic background accounts for approximately 50-80% of the variability in pubertal onset and progression [129]. Certain ethnic groups, notably African American and Hispanic populations, exhibit an earlier onset of puberty attributed to genetic and nutritional factors [130]. Environmental factors, including substances with the potential to interfere with the endocrine system (such as phthalates, dioxins, polybrominated biphenyls, and polychlorinated biphenyls), appear to play a role in influencing the timing of puberty [129,131,132]. In future research investigating the link between gut microbiota and precocious puberty, as well as its underlying mechanisms, it is crucial to meticulously address and control for the impact of these confounding variables.” (page 10, lines 434-445).
Comment 9: The review includes a variety of study types (observational studies, animal experiments) which inherently have different levels of evidence. The authors could provide a more detailed discussion of how this heterogeneity affects the overall conclusions.
Response: Thank you for your suggestion. We revised the conclusion reads: “Present studies predominantly adopt a cross-sectional approach, yielding correlations with limited persuasiveness. The imperative necessity lies in conducting longitudinal cohort studies, offering more robust evidence with a causal direction. These studies are crucial to comprehensively investigate the dynamics of the gut microbiota and its potential involvement in the development of precocious puberty in humans. Additionally, animal experiments can validate whether the gut microbiota can regulate precocious puberty and elucidate the molecular mechanisms involved. Nevertheless, evidence from animal studies may not precisely replicate human physiology and pathology. Therefore, clinical trials are imperative to elucidate the causal association between the gut microbiota and precocious puberty in humans.” (page 15, lines 523-532).
Comment 10: Regarding probiotics, the shift in the gut microbiota may be transient and temporary (citation: pubmed.ncbi.nlm.nih.gov/36986088) as several treatment trials for probiotics have failed to find significant alterations in gut microbiome; individuals may require longer duration of treatment to have therapeutic effects.
Response: Thank you for your suggestion. We added the characteristic of probiotics therapy and propose the advice of it reads: “However, concerning probiotic therapy, the alterations in the gut microbiota it induces may be transient [139]. Several probiotic treatment trials have failed to observe significant changes in the gut microbiota, suggesting that individuals might require an extended treatment duration to achieve therapeutic effects. Future clinical trial re-search should place greater emphasis on post-intervention follow-ups to assess the optimal intervention duration for probiotic therapy.” (page 11, line 469-474). We also cited the study you mentioned to support the insight (citation: pubmed.ncbi.nlm.nih.gov/36986088) (page 11, line 470).

Reviewer 2 Report
Comments and Suggestions for Authors
Page 2, line 76: The term "mode" is repeated.
Figure 1: Modify the label in the box describing gut microbiota functions. Change "Metabolic" to "Nutrients metabolism" and "Communicate with CNS" to "gut brain axis communication."
Figure 2: Enhance the descriptions of the interactions between gut microbiota and the HPGA, providing detailed information on each interaction.
In section 4, titled "Interaction and Potential Mechanisms between Gut Microbiota and Precocious Puberty": Remove the mention of "Probiotics" as an interaction in section 4.5. Instead, include a new subsection discussing possible therapeutics based on gut microbiota for precocious puberty. This subsection should incorporate information on the use of probiotics in precocious puberty and explore the potential of other therapies such as fecal microbiota transplant.
In the conclusion section, on page 13, lines 475-477: Revise the sentence regarding probiotics treatment as an "interaction between gut microbiota and precocious puberty." Instead, rephrase it to indicate that probiotics treatment represents a potential therapeutic approach for precocious puberty.
Author Response
Response to Reviewer2
The authors would like to thank the reviewer for your helpful comments. We feel these comments have strengthened the manuscript considerably. The changes/revisions we made in the revised manuscript are highlighted in yellow color.
Reviewer #2:
Comment 1: Page 2, line 76: The term "mode" is repeated.
Response: Thank you for your detailed review. We deleted the word “mode” on page 2, line 76.
Comment 2: Figure 1: Modify the label in the box describing gut microbiota functions. Change "Metabolic" to "Nutrients metabolism" and "Communicate with CNS" to "gut brain axis communication."
Response: Thank you for your detailed review. We revised the label in the box describing gut microbiota functions reads: “Nutrients metabolism; Modulate host energy balance; Regulate inflammatory immune responses; Enhance intestinal barrier function; Gut brain axis communication.” (page 3, Figure 1).
Comment 3: Figure 2: Enhance the descriptions of the interactions between gut microbiota and the HPGA, providing detailed information on each interaction.
Response: Thank you for your suggestion. We revised the descriptions of the interactions between gut microbiota and the HPGA on Figure 2 reads: “Metabolite: serotonin, dopamine, nitric oxide, SCFAs, bile acids; Nutritional status: overnutrition, energy imbalance, early nutrition; Hormone regulation: LH, FSH, GnRH, estrogen, insulin, androgens.” (page 6, Figure 2).
Comment 4: In section 4, titled "Interaction and Potential Mechanisms between Gut Microbiota and Precocious Puberty": Remove the mention of "Probiotics" as an interaction in section 4.5. Instead, include a new subsection discussing possible therapeutics based on gut microbiota for precocious puberty. This subsection should incorporate information on the use of probiotics in precocious puberty and explore the potential of other therapies such as fecal microbiota transplant.
Response: Thank you for your review. We removed the mention of "Probiotics" as an interaction in section 4.5. Instead, we added a section exploring the prospects of microbiota-associated therapies reads: “
- The Prospects of Microbiota-Associated Therapies
6.1. Probiotics
Probiotics have been widely used in recent years for the prevention and adjunctive therapy of various diseases due to their non-invasive nature [133-136]. Regarding probiotic-related research in precocious puberty, there are currently only two animal experimental studies that have been reported. One study reported that probiotic treatment could reverse soy isoflavone (SI)-induced precocious puberty in female mice, possibly due to increased production of SCFAs brought about by changes in the gut microbiota of the recipient mice [137]. The probiotics used in this study were commercial, consisting of live Bifidobacterium longum, Lactobacillus bulgaricus, and Streptococcus thermophilus [137]. Another animal experiment demonstrated that probiotic intake could reverse early maternal separation stress-induced precocious puberty in female rats, indicating that probiotic therapy restored the normal onset of puberty in rodents [138]. The probiotics used in this study were also commercial, with ingredients including 95% Lactobacillus rhamnosus R0011 and 5% Lactobacillus helveticus R0052 [138]. These studies all suggest that probiotic intake has a stabilizing effect on the timing of puberty onset in rodents, although the specific mechanisms need further exploration. Furthermore, the translation of the aforementioned animal study results to the human population requires additional clinical trial evidence. Future research efforts should prioritize more population-based intervention studies. Simultaneously, current re-search on probiotics for precocious puberty lacks specificity, and future research should further investigate probiotics specifically related to precocious puberty for more targeted and effective treatment.
However, concerning probiotic therapy, the alterations in the gut microbiota it induces may be transient [139]. Several probiotic treatment trials have failed to observe significant changes in the gut microbiota, suggesting that individuals might require an extended treatment duration to achieve therapeutic effects. Future clinical trial re-search should place greater emphasis on post-intervention follow-ups to assess the optimal intervention duration for probiotic therapy.
6.2. Fecal Microbiota Transplantation (FMT)
FMT refers to the transfer of fecal material containing the gut microbiota from a healthy donor to a recipient with dysbiosis, using methods such as colonic infusion, nasogastric, nasoenteric, or endoscopic approaches. The objective is to restore the normal diversity and functionality of the gut microbiota [140,141]. Currently recognized as an effective treatment for recurrent Clostridium difficile infection [142], FMT is also considered a potential therapy for certain extraintestinal diseases, including neurodegenerative disorders, owing to the bidirectional communication of the gut-brain axis[143,144]. Additionally, recent animal experimental evidence suggests FMT as a potential intervention for anti-aging and altering the life course stages[145]. This raises the question of whether FMT could also influence the timing of puberty, a crucial life stage, providing a novel direction for the treatment of precocious puberty. However, no studies have provided clues to date, and further research is warranted to explore the impact of FMT on the onset of puberty in humans.” (pages 10-11, lines 445-487).
Comment 5: In the conclusion section, on page 13, lines 475-477: Revise the sentence regarding probiotics treatment as an "interaction between gut microbiota and precocious puberty." Instead, rephrase it to indicate that probiotics treatment represents a potential therapeutic approach for precocious puberty.
Response: Thank you for your review. We revised the sentence in the conclusion regarding probiotics treatment as an "interaction between gut microbiota and precocious puberty." as to: “Building upon prior research, this paper suggests several novel avenues for investigating the interaction between gut microbiota and precocious puberty. First, the interaction between the gut microbiota and precocious puberty mainly occurs through four pathways: microbiota metabolism, microbiota-hormone interactions, microbio-ta-nutritional status, and microbiota-HPGA interactions. These pathways are closely related, and future research should combine nutritional status, microbiota metabolism products, hormone regulation, and HPGA interactions to systematically elucidate the potential homeostatic system between the gut microbiota and precocious puberty. This will further enhance our understanding of the pathological mechanisms of precocious puberty. Secondly, probiotics and fecal microbiota transplantation (FMT) emerge as potential non-invasive treatments for precocious puberty. While the application of probiotics in treating precocious puberty is currently limited to animal experiments, we posit that future investigations should prioritize additional clinical experiments to further explore the therapeutic potential of probiotics for precocious puberty.” (page 15, lines 532-545).

Round 2
Reviewer 1 Report
Comments and Suggestions for Authors
1. The structure of the paper still needs to be reworked. It needs to be better structured as there are still many long paragraphs and it goes between (sometimes unrelated) issues. The definition of precocious puberty should also be introduced from the outset rather than only at section 4.1
2. It is not standard practice to put the entire study title in Table 1 and the key findings should be further summarised and made more concise please.
3. Same as the above, there is no need for the full study titles to be displayed in Table 2, just first author and year would suffice.
4. The limitations of the study should be placed in the discussion rather than right at the end of the paper in the conclusion section.
5. "The imperative necessity" - redundant to say imperative and necessary in the same phrase.
Comments on the Quality of English LanguageModerate edits required.
Author Response
Response to Reviewer 1
The authors would like to thank the reviewer for your helpful comments. We feel these comments have strengthened the manuscript considerably. The changes/revisions we made in the revised manuscript are highlighted in yellow color.
Reviewer #1:
Comment 1: The structure of the paper still needs to be reworked. It needs to be better structured as there are still many long paragraphs and it goes between (sometimes unrelated) issues. The definition of precocious puberty should also be introduced from the outset rather than only at section 4.1
Response: Thank you for your detailed review. In order to make the structure of the paper clearer, we added several subheadings according to the content the paper reads: “4.3. Etiology and Risks of Precocious Puberty; 5.1.1 Neurotransmitter metabolic pathways; 5.1.2 Amino acid metabolic pathways; 5.1.3 Lipid metabolism; 5.1.4 Bile acid metabolism.” (page 4, line 168; page 7, line 265; page 7, line 274; page 7, line 282; page 8, line 306). In addition, the definition of precocious puberty has been introduced in introduction of the manuscript (page 2, lines 45-46).
Comment 2: It is not standard practice to put the entire study title in Table 1 and the key findings should be further summarized and made more concise please.
Response: Thank you for your constructive comment. We removed the fully study titles displayed in Table 1 and remained author and year of review studies (pages 11-13, Table 1). We also further summarized the key findings of review studies to make more concise (pages 11-13, Table 1).
Comment 3: Same as the above, there is no need for the full study titles to be displayed in Table 2, just first author and year would suffice.
Response: Thank you for your detailed review. We removed the fully study titles displayed in Table 2 and remained author and year of review studies (page 13, Table 2). We also further summarized the key findings of review studies to make more concise (page 13, Table 2).
Comment 4: The limitations of the study should be placed in the discussion rather than right at the end of the paper in the conclusion section.
Response: Thank you for your review. We added the discussion reads: “The exact mechanisms underlying precocious puberty remain incompletely understood. Identified causes include congenital and acquired central nervous system damage, genetic changes, environmental endocrine disruptors, and premature expo-sure to sex hormones. However, the etiology of CPP remains unclear in 74%-90% of affected girls, making it the most common form of precocious puberty [9,11,12]. The significant changes in the gut microbiota before and after puberty suggest a relationship with sexual maturation. Moreover, increasing observational studies indicate disturbances in the gut microbiota of children with precocious puberty. Non-invasive treatments such as probiotics and fecal microbiota transplantation are increasingly being used in clinical practice to adjunctive treat gastrointestinal diseases [150] (such as ulcerative colitis, Crohn's disease, irritable bowel syndrome, etc.), brain-gut axis-related neurological diseases [151-153] (such as Parkinson's disease, Alzheimer's disease, anxiety, depression, autism, etc.), endocrine system and immune system diseases [154,155] (such as diabetes, obesity, lupus, rheumatoid arthritis, allergies, etc.). This suggests that the mechanisms by which precocious puberty interacts with the gut microbiota is a promising research area, providing a theoretical basis for the development of specific probiotic strains or fecal microbiota transplantation treatments for precocious puberty.
However, it is currently unknown whether the gut microbiota can affect precocious puberty. Present studies predominantly adopt a cross-sectional approach, yielding correlations with limited persuasiveness. The necessity lies in conducting longitudinal cohort studies, offering more robust evidence with a causal direction. These studies are crucial to comprehensively investigate the dynamics of the gut microbiota and its potential involvement in the development of precocious puberty in humans. Additionally, animal experiments can validate whether the gut microbiota can regulate precocious puberty and elucidate the molecular mechanisms involved. Nevertheless, evidence from animal studies may not precisely replicate human physiology and pathology. Therefore, clinical trials are imperative to elucidate the causal association between the gut microbiota and precocious puberty in humans.
Building upon prior research, this paper suggests several novel avenues for investigating the interaction between gut microbiota and precocious puberty. First, the interaction between the gut microbiota and precocious puberty mainly occurs through four pathways: microbiota metabolism, microbiota-hormone interactions, microbiota-nutritional status, and microbiota-HPGA interactions. These pathways are closely related, and future research should combine nutritional status, microbiota metabolism products, hormone regulation, and HPGA interactions to systematically elucidate the potential homeostatic system between the gut microbiota and precocious puberty. This will further enhance our understanding of the pathological mechanisms of precocious puberty. Secondly, probiotics and fecal microbiota transplantation (FMT) emerge as potential non-invasive treatments for precocious puberty. While the application of probiotics in treating precocious puberty is currently limited to animal experiments, we posit that future investigations should prioritize additional clinical experiments to further explore the therapeutic potential of probiotics for precocious puberty.
However, we also realized the limitation of this study. This narrative review, lacking specific criteria for evaluating the quality of population studies, has subjective inclusion and exclusion criteria for literature. Therefore, in the future, there should be an effort to conduct systematic reviews in this field to the extent possible.” (pages 13-14, lines 508-555).
Additionally, we revised the conclusion reads: “In conclusion, current research preliminarily confirms a correlation between precocious puberty in humans and the gut microbiota. Animal studies have shown that specific gut microbiota and their metabolites can reverse precocious puberty in rodent models. However, the causal effects and underlying interaction mechanisms between human precocious puberty and gut microbiota remain to be elucidated. This narrative review summarizes the potential molecular mechanisms mentioned in existing studies and proposes potential microbiome-related therapeutic approaches for precocious puberty. Future population studies are needed to clarify the causal relationship between the gut microbiota and human precocious puberty, as well as to elucidate their potential interaction mechanisms. Concurrently, clinical trials exploring specific probiotics and their metabolites for children with precocious puberty could provide new insights for non-invasive treatment options.” (pages 14-15, lines 556-568).
Comment 5: "The imperative necessity" - redundant to say imperative and necessary in the same phrase.
Response: Thank you for your suggestion. We revised the sentence reads: “The necessity lies in conducting longitudinal cohort studies, offering more robust evidence with a causal direction.” (page 14, lines 530-531).
Additionally, we have revised the English language in this paper to ensure readability.

Reviewer 2 Report
Comments and Suggestions for Authors
Page 10, Line 438. After the sentence "Certain ethnic groups, notably African American and Hispanic populations, exhibit an earlier onset of puberty attributed to genetic and nutritional factors [130]," it is noteworthy to mention that genetic factors can impact the gut microbiome in humans. Numerous literature sources have documented this association.
Author Response
Response to Reviewer2
The authors would like to thank the reviewer for your helpful comments. We feel these comments have strengthened the manuscript considerably. The changes/revisions we made in the revised manuscript are highlighted in yellow color.
Reviewer #2:
Comment 1: Page 10, Line 438. After the sentence "Certain ethnic groups, notably African American and Hispanic populations, exhibit an earlier onset of puberty attributed to genetic and nutritional factors [130]," it is noteworthy to mention that genetic factors can impact the gut microbiome in humans. Numerous literature sources have documented this association.
Response: Thank you for your detailed review. We added the genetic factors can impact the gut microbiome in humans and cited the relevant studies reads: “Concurrently, a substantial body of research indicates that genetic factors can also in-fluence the human gut microbiome [131-133]. Host genetic polymorphisms and/or mutations progressively alter the gut microbiota, intersecting with other environmental influences, leading to changes in the composition and function of the gut microbial community [134].” (page 10, lines 441-445).
